# Optic Nerve Sheath Diameter: A Cross-Sectional Study of Ultrasonographic Measurement in Healthy Black South African Adults

**DOI:** 10.3390/life13101979

**Published:** 2023-09-28

**Authors:** Lindy Trollip, Kerry Alberto, Aubrey Makgotloe

**Affiliations:** Department of Ophthalmology, University of Witwatersrand, Johannesburg 2193, South Africa; kerryalberto@yahoo.com (K.A.); aubrey.makgotloe@wits.ac.za (A.M.)

**Keywords:** optic nerve sheath diameter, optic nerve ultrasound, ONSD, ocular ultrasonography, raised intracranial pressure

## Abstract

Ultrasonographic optic nerve sheath diameter (ONSD) measurement is an accurate, portable, and non-invasive method of detecting raised intracranial pressure that can also reflect dynamic, real-time changes in intracranial pressure fluctuations. Various studies have shown the mean range of ONSD to vary greatly across different population groups. This study aimed to determine the mean ONSD in healthy Black South African adults. In this cross-sectional study, healthy black South African adult participants underwent optic nerve sheath ultrasound of the right eye, with the diameter being measured at 3 mm behind the retina in two different planes. The average of the two measurements was used to find the mean optic nerve sheath diameter. This measurement was compared to that found in a Canadian adult population, and the effect of age, gender, and co-morbidities on ONSD was assessed. A total of 99 participants were included in this study, of which 39 were male and 60 were female. The mean ONSD was found to be 5.1 mm (SD ± 0.33). This value was significantly higher than the mean ONSD observed in the Canadian population (*p* < 0.001). There was no significant difference found between the mean ONSD in males and females (*p* = 0.652), and both age and presence of co-morbidities were not found to significantly correlate with ONSD. (*p* = 0.693 and *p* = 0.974, respectively).

## 1. Introduction

Raised intracranial pressure (ICP) is a common presenting complaint to emergency departments and ophthalmology departments around the world. If it is left undiagnosed or untreated, it can lead to significant cerebral hypoxia, permanent disability, or even death. In South Africa, the incidence of traumatic brain injury is estimated to be around 316 per 100,000 per year, which is higher than the estimated incidence of 90.5 per 100,000 population in the United States of America and 200 per 100,000 population in Europe [1]. Raised intracranial pressure can be detected both radiographically, using Computed Tomography (CT) or Magnetic Resonance Imaging (MRI), and ultrasonographically by measuring the distension of the optic nerve sheath surrounding the optic nerve.

The optic nerve is formed from the retinal nerve fibre layer of the retina. As the optic nerve exits the globe posteriorly, it becomes encased by the dura, arachnoid, and pia mater, forming the optic nerve sheath, and travels via the orbit and optic canal to the optic chiasm. Therefore, intracranial subarachnoid fluid can travel freely between the intracranial subarachnoid space and the perineural subarachnoid space surrounding the optic nerve. Fluctuations in intracranial pressure can be measured by assessing the distension of the optic nerve sheath, which is believed to be the most distensible at 3 mm behind the globe, with a maximum distension of up to 7.5 mm. The distension then reaches a plateau at 7.5 mm despite any further rise in intracranial pressure [2].

Ultrasonographic measurement of the optic nerve sheath diameter has been found in multiple studies to be a specific and sensitive predictor of raised intracranial pressure [3,4,5,6], and many emergency departments around the world have adopted this non-invasive, bedside investigation as a screening tool for detecting this condition. The use of optic nerve ultrasound has also been more recently investigated in a broader range of ophthalmic conditions. This includes its use in the initial screening of idiopathic intracranial hypertension (IIH) and subsequently monitoring the response of intracranial pressure to the treatment prescribed by performing serial optic nerve sheath diameter measurements [7]. It is also being used for the detection of other optic nerve pathologies, including normal tension glaucoma [8,9] and both inflammatory and ischaemic optic neuritis [10,11].

However, various published studies [12,13,14,15,16] conducted globally have shown the mean and range of optic nerve sheath diameter to differ vastly across different population groups, with a mean optic nerve sheath diameter found to vary from 3.68 mm (95% confidence interval 2.85 to 4.40 mm) in a Canadian population group [13] to 5.1 mm (95% confidence interval 4.7 to 5.4 mm) in a Chinese population group [15]. These variations in the reported mean values highlight the importance of finding a normative database within different population groups for this tool to be accurately used in both screening and diagnosis of various pathological conditions.

The aim of our study was to determine the mean optic nerve sheath diameter in a healthy black South African adult population and to assess if this measurement differed significantly from that found in a Canadian population [13]. The effect of gender, age and co-morbidities on optic nerve sheath diameter was also assessed.

## 2. Materials and Methods

This study was a descriptive cross-sectional study conducted at Helen Joseph Hospital, a tertiary-level hospital affiliated with the University of the Witwatersrand in Johannesburg, South Africa. Healthy adult volunteers in the Emergency Department, including adult members of the hospital staff and patients’ relatives, were invited to participate in this study. In total, 105 adult volunteers were enrolled in the study between March and December 2022. All of the participants were above the age of eighteen years and were self-reported South Africans of black African race. Potential participants were excluded if they reported any clinical history in keeping with optic nerve pathology or brain-related neurological pathology—both current or previous. For this study, the right eye of every participant was used. The volunteers underwent slit lamp biomicroscopic examination to assess the fundoscopic appearance of the optic nerve head. Intraocular pressure (IOP) measurements were performed using a Perkins applanation tonometer (PAT) of the right eye in each patient. Subjects were included in the study if both the optic nerve examination and IOP were found to be within normal limits as defined subsequently.

Normal fundoscopic optic nerve appearance [17] was classified as a pink neuroretinal rim following the ISNT (Inferior, Superior, Nasal, Temporal) rule, a cup-to-optic disc ratio of less than or equal to 0.4, clear optic disc margins and the absence of optic disc pathology. If a patient had a cup-to-optic disc ratio of 0.5 or greater, but otherwise normal ophthalmoscopic appearance, they were included in the study, provided that they had a normal optical coherence tomography retinal nerve fibre layer (OCT RNFL) on Copernicus Revo Spectral Domain Optical Coherence Tomography (SD-OCT, Optopol Technology Zawiercie, Polska. The OCT RNFL scan required a quality index of greater than or equal to 7 to be deemed acceptable. An intraocular pressure of between 8 and 21 mmHg was considered to be within normal limits. All volunteers willing to participate in the study gave written informed consent, and the study was approved by the Human Research Ethics Committee of the University of the Witwatersrand (ethics clearance number M191159) and adhered to the tenets of the Declaration of Helsinki.

At study entry, data collected included age, gender, race, known co-morbidities, optic nerve appearance, intraocular pressure values, and two optic nerve sheath measurements of the right eye. The ultrasonography was conducted manually by a single ophthalmology registrar with both previous experience and training in optic nerve sheath ultrasound. A 10 MHz linear probe, on a Mindray M7 premium ultrasound machine (Mindray UK Limited) in B scan mode, was used to take axial cross-sectional measurements of the optic nerve sheath diameter in the right eye of all the study participants. The participants were placed in a relaxed supine position with head and upper body elevation of approximately 30 degrees and the probe was gently placed over a closed eyelid in a superolateral position. Generous coupling gel was used to alleviate pressure on the ocular contents. Participants were asked to keep their eyes in primary gaze to minimize motion artifact and this was achieved by instructing them to maintain focus on a fixed distant object directly in front of them with the open left eye. Frozen images of the optic nerve were captured, and the measurement of the optic nerve sheath diameter was taken using electronic calipers at 3 mm behind the retina (Figure 1). The measurement was taken between the external borders of the hyperechoic optic nerve sheath. Each optic nerve sheath was measured twice, with the probe orientated in two different planes at least thirty degrees apart, and the average of the two measurements was taken to ensure improved accuracy.

Optic nerve sheath measurement images were then reviewed by a consultant ophthalmologist and participants were excluded if the image and measurements were not assessed to be clear and/or accurate by the consultant ophthalmologist. Four of the study participants were excluded after a review of the images and measurements by the consultant ophthalmologist, and a further two participants were excluded for failing to have a normal retinal nerve fibre layer after large cup-to-optic disc ratios were noted on fundoscopic examination. One participant had a cup-to-optic disc ratio (CDR) of 0.5 and a normal RNFL and, therefore, met inclusion criteria and was included in the study. This led to a total number of 99 participants that were included for data analysis. (Figure 2). All data were captured on Microsoft Excel in a password-protected computer and exported to SPSS version 27 for statistical analysis.

The mean optic nerve sheath diameter found in our population was compared to that found in a Canadian study by Goeres et al. on 120 healthy adult volunteers [13].

### Statistical Analysis

A required sample size of ninety-seven participants was calculated based on a standard deviation of 0.5 [13], 0.1 mm margin of error and five percent of type one error using the Canadian study by Goeres et al. as a guide for the sample size calculation [13,18].

Kolmogorov–Smirnov testing for normality indicated that intraocular pressure, both the first and second optic nerve sheath diameter measurements and the mean optic nerve sheath diameter were all normally distributed. Descriptive statistics were calculated for the ONSD including mean, standard deviation (SD), minimum, maximum and 95th percentile for quantitative data, and qualitative data were expressed using frequency and percentage. To test for the reliability of the repeated ONSD measurements within individuals, Intraclass Correlation Coefficients (ICC)—a two-way mixed model was used for absolute agreement. Independent *t*-test was used to test for differences across gender and co-morbidities and linear regression was used to assess the effect of age on ONSD after correcting for gender, and the effect of gender on ONSD after correcting for age. A one-sample *t*-test was used to compare our study finding of mean ONSD to the mean ONSD that was found in a Canadian population [13]. A *p*-value of 0.05 was used for statistical significance. SPSS version 27 was used for statistical analysis. The costs of the study were provided by the corresponding author, and, therefore, there was no cost to the participants and no external funding received.

## 3. Results

### 3.1. Demographic Data

A total of 99 participants were included in the statistical analysis, with 60 females and 39 males. The age ranged from 20 to 64 years. A total of 79 participants had no known history of co-morbidities, while 20 participants reported co-morbidities which included hypertension, diabetes mellitus, asthma, and Human Immunodeficiency Virus (HIV). The demographic data of the study participants are summarized in Table 1:

### 3.2. ONSD Measurements

The range of ONSD found in our adult population group was 4.3 mm to 6.0 mm. The mean measurements for all ONSD in the participants are summarized below in Table 2. The mean ONSD was found to be 5.1 mm (SD ± 0.33) with a 95th percentile of 5.6 mm. The ICC analysis for the reliability of the ONSD measurements was 0.94 (95% confidence interval 0.900–0.956), therefore showing acceptance for reliability.

A Bland–Altman plot was used to assess intraobserver variability between the two measurements performed in each participant (Figure 3).

### 3.3. Gender and Age

When assessing the effect of gender on ONSD, there was no significant difference found between the mean ONSD in males and females (5.07 mm vs. 5.10 mm, respectively, *p* = 0.652). Linear regression analysis showed that age and gender account for 3% (R^2^ = 0.030) of the variance in ONSD, (F (2, 96) = 1.465, *p* = 0.236). When included together as independent variables, neither age (β = 0.166, *p* = 0.102) nor gender (β = 0.040, *p* = 0.693) were found to be significant predictors of ONSD.

### 3.4. Co-Morbidities

The mean ONSD in those participants who reported one or more co-morbidities was 5.1 mm and 5.1 mm in those who reported no known co-morbidities. This was not found to be a significant difference (*p* = 0.974).

### 3.5. Comparison to a Canadian Population

The mean ONSD in a Canadian population was found to be 3.68 mm (95% confidence interval 2.85–4.40 mm). This was found to be significantly different from the mean of 5.1 mm found in our population group when analysed using a one-sample *t*-test (*p* < 0.001).

## 4. Discussion

The optic nerve sheath diameter can be measured using various methods, including a Computerized Tomography scan, Magnetic Resonance Imaging scan, and ultrasound imaging. Direct measurements of the intracranial pressure are carried out using invasive methods, including intraventricular catheters and intraparenchymal probes. Ultrasonographic measurement of the ONSD has the benefit of being a relatively quick, non-invasive, and portable test which can spare the patient and hospital costly and time-consuming radiological investigations. It has also been found to accurately detect dynamic, real-time changes in intracranial pressure fluctuations when the optic nerve sheath diameter is assessed before and after lumbar puncture for intracranial pressure measurements [19]. In resource-limited settings, including rural hospitals in South Africa, where radiological investigations are not easily available, the expanded use of ultrasonographic screening for raised intracranial pressure and other pathological conditions is especially appealing. The use of transorbital ultrasonographic measurement of ONSD also has the added benefit of not requiring a skilled ultrasonographer to accurately measure the diameter. A study by Potgieter et al. [20] showed that novice operators can be taught in a single supervised training session to measure ultrasonographic optic nerve sheath diameters in healthy volunteers, with precision and accuracy comparable to an experienced ultrasonographer. This makes it a valuable tool to use in clinics and hospitals around the world. However, the lack of consensus for the cut-off value of the upper limit of normal ONSD and the paucity of studies in black African populations is what led us to conduct this study to find a normative database in this population group that would enable us to accurately use this investigation going forward.

Our study shows that in a healthy black South African adult population, the mean ONSD is 5.1 mm (SD ± 0.33). The 95th percentile was 5.6 mm, and the range of optic nerve sheath diameter was found to be 4.3 mm to 6.0 mm. The mean optic nerve sheath diameter in our study population was found to be significantly different from the mean optic nerve sheath diameter of 3.68 mm (95% CI 2.85–4.40 mm) found in a Canadian population group [11] (*p* < 0.001) and similar to that found in a Chinese population group of 5.1 mm (95% CI 4.7–5.4 mm)) [13]. There was no significant correlation found between age or gender on ONSD and this is consistent with other studies published [15,16,21,22]. As we tested only the right eye of every participant of this study, we could not identify if there was any difference in optic nerve sheath diameter found between the right and left eye of each participant; however, other studies have not found a significant difference between the two eyes when bilateral optic nerve sheath diameters were analysed [12,15,21].

The co-morbidities noted in our participants included hypertension, diabetes, HIV/AIDS, and Asthma. We did not find any significant correlation between the presence of comorbidities and mean ONSD (*p* = 0.974) when we assessed the difference between the mean ONSD in those who reported one or more comorbidities and those who did not have any reported comorbidities. It is important to note that our study was not powered to assess the correlation of the individual comorbidities to ONSD and, therefore, the correlation between individual comorbidities on ONSD cannot be reported from this study. A study carried out by Ebisike et al. looking at ONSD in an adult Nigerian population suggested that HIV/AIDS, including the treatment with Highly Active Antiretroviral Therapy (HAART), may lead to an increase in the optic nerve diameter and optic nerve sheath diameter in patients with this condition, notably in females [23]. Although our study assessed the correlation between the presence of co-morbidities (including HIV/AIDS) and ONSD, we did not have enough participants with HIV/AIDS to assess for a statistically significant effect on ONSD and this would need to be carried out in further studies to correlate the effect of HIV/AIDS on ONSD in comparison to HIV negative controls in our population due to the significant burden of HIV/AIDS.

The results of our study add to the variation in mean ONSD found in various population groups across the globe and there have been various reasons suggested as to the variation found in mean ONSD in these different studies. The first of these reasons suggested is the variation in the method of ONSD measurement used [22,24]. The optic nerve sheath is measured as the area between the external borders of the hyperechoic optic nerve sheath and the arachnoid mater. If the measurement is taken from the internal borders of the hyperechoic sheath, then this measurement is of the optic nerve itself and will lead to an underestimation of the mean ONSD. On the contrary, if the measurement includes the hypoechoic area external to the optic nerve sheath, then this measurement will include the dura mater, retrobulbar fat, and artifact surrounding the optic nerve and will lead to overestimation of the ONSD. The measurement is taken at three millimetres behind the globe as this is where the optic nerve is believed to be the most distensible and, therefore, the most sensitive to fluctuations in intracranial pressure [25]. To increase the accuracy of the nerve sheath measurement, various methodologies have been suggested, including colour Doppler imaging to look for the central retinal vessels thereby confirming the course of the optic nerve to avoid measuring artifacts [25,26] and the use of an Amplitude scan (A-scan) ultrasonography to avoid the “blooming effect” when Brightness scan (B- scan) mode of the ultrasound machine is used without a standardized gain setting [27]. High-frequency ultrasound elastography, an emerging imaging modality, has been found to detect subtle changes in ocular structures before these changes are evident on B scan ultrasonography but further research is required to determine if this could be a useful tool in optic nerve sheath imaging [28].

Another suggestion for the variation in optic nerve sheath diameter is ethnic/racial/genetic heterogeneity. When searching the literature published on racial variations in ocular anatomy, we found studies published on a black South African population suggesting racial variation in normative ocular anatomy. This included the finding that black South African adults have higher intraocular pressures and thinner central corneal thickness (CCT) values than other ethnic groups studied [29]. In relation to the optic nerve, black South Africans have also been found to have significantly thicker retinal nerve fibre layers as imaged on OCT when compared to the European database that has been determined by the manufacturer of that OCT machine [30].

Although not representing a South African population, there are also studies published on an international population showing ethnic differences in other ocular structures, including anterior chamber depth and anterior chamber angle structure [31]. Regarding the optic nerve head, black persons were found to have an optic disc area twenty-six percent greater than caucasian persons [32] and a significantly larger vertical disc diameter reading on optic disc imaging [33].

Both the racial and methodical variations suggested above may account for the variation in mean ONSD, and this further emphasizes the need for a population-based normative database and a standardized protocol for optic nerve sheath ultrasonography. Accurate use of this tool will ultimately aid in optimizing our management and monitoring of the patients in our care for a broad range of medical conditions.

The strengths of this study include this being the first study, to our knowledge, looking at determining the mean ONSD in healthy black South African adults. Although the majority of studies found during the literature review did not include an ocular examination on the participants undergoing optic nerve sheath ultrasonography, we included optic nerve head examination on slit lamp biomicroscopy and intraocular pressure measurements by applanation tonometry to further exclude any possible optic nerve pathologies that may confound the data collection for a normative database. To increase the accuracy of the optic nerve sheath diameter measurement and avoid artifact, the optic nerve sheath was measured in two different cross-sectional planes at least thirty degrees apart and the average of the two readings was used for each participant. We hope that this finding of the normal range of ONSD and the upper limit cut-off value can be used in subsequent studies to look at the use of ONSD in detecting and monitoring other pathological conditions of the optic nerve, including IIH and optic neuritis.

There are a few limitations of this study to be noted. Firstly, the optic nerve sheath measurements were taken by a single investigator—although experienced in optic nerve ultrasound and trained in level one ultrasonography and then captured images and measurements reviewed by a second ophthalmologist. The study could have been strengthened through the inclusion of a second investigator, with a masking of measurements between the two investigators, to decrease the risk of observer bias and then subsequently comparing these for interobserver variability. The ultrasound machine used in our study differed from that used in the study by Goeres et al. [13] and, therefore, using different ultrasound manufacturers could also be considered in future studies to assess for inter-device variability that may account for some of the variation in mean ONSD measurements reported in different studies. Due to this study being conducted in the emergency department and outside normal clinic hours, only participants with a cup-to-optic disc ratio greater than 0.4 on fundoscopy required a normal OCT RNFL before inclusion into data analysis but this could be performed on all study participants in future studies to further strengthen the assessment of the optic nerve in excluding any subclinical optic neuropathy. Lastly, our study was not powered to determine the effect of individual comorbidities on optic nerve sheath diameter, and this would be a useful consideration for further studies.

## 5. Conclusions

Optic nerve sheath ultrasound is a quick and cost-effective non-invasive, bedside tool used to accurately assess for both raised intracranial pressure and optic nerve-related pathology. It has been found to be highly sensitive and specific at detecting changes in intracranial pressure but requires population-based studies to determine a cut-off value for accurate use due to the vast heterogeneity in the normal range of optic nerve sheath diameter values found in studies published across the world. Our study conducted on healthy black South African adult volunteers found the mean optic nerve sheath diameter to be 5.1 mm (SD ± 0.33). ONSD was found to be independent of gender, age, and the presence of co-morbidities. We suggest an upper limit cut-off value of 5.6 mm be used when assessing ONSD in Black South African adults for accurate screening of raised intracranial pressure.

## Figures and Tables

**Figure 1 life-13-01979-f001:**
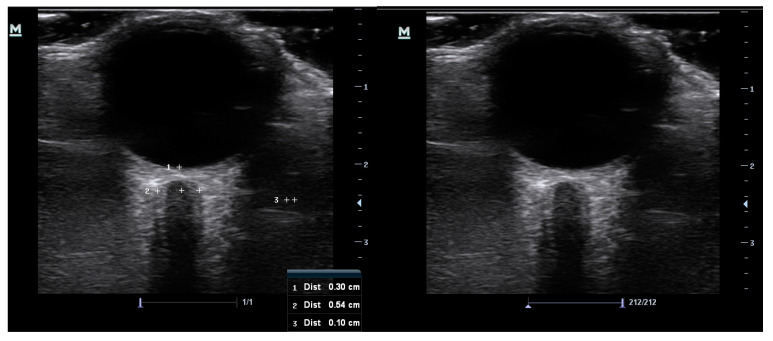
Images captured of the optic nerve sheath ultrasonography before measurement (**left**) and showing the measurement of the sheath at 3 mm behind the retina (**right**). The hyperechoic sheath surrounds the optic nerve at the centre.

**Figure 2 life-13-01979-f002:**
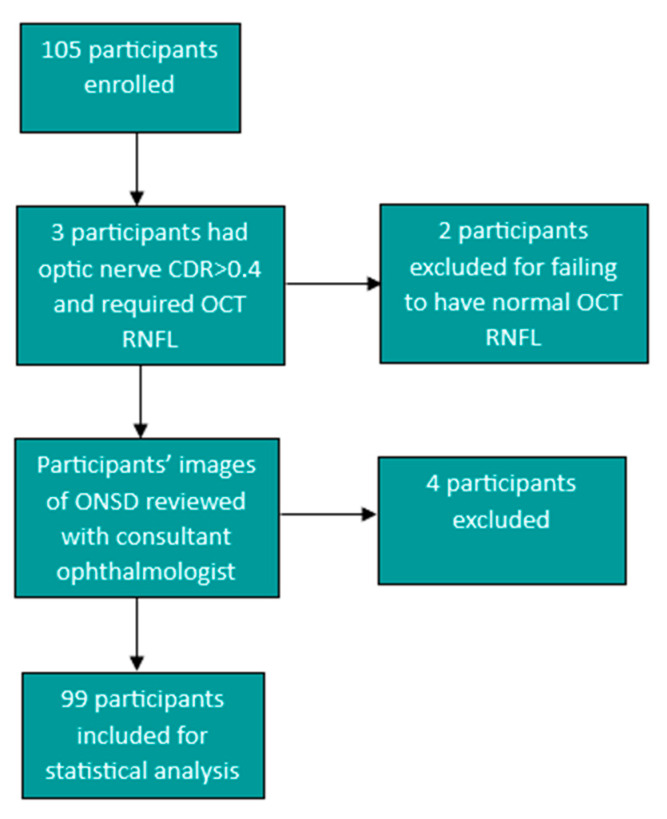
Flow diagram showing the participants excluded from the study to give the total of 99 participants for statistical analysis.

**Figure 3 life-13-01979-f003:**
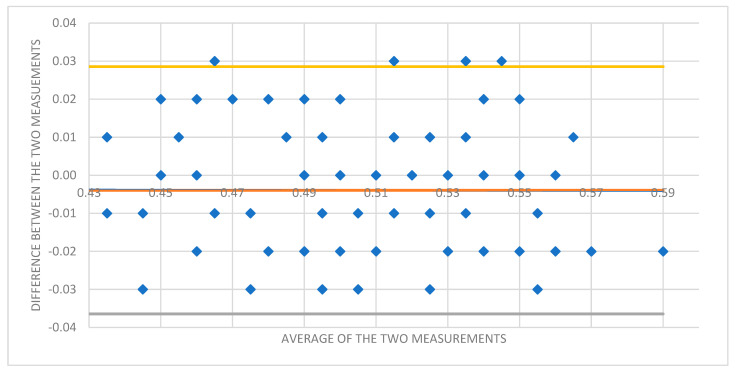
Bland–Altman plot showing the difference between ONSD 1 and ONSD 2 measurements for each participant to assess for intraobserver variability.

**Table 1 life-13-01979-t001:** Summary of enrolled participants’ demographic data.

Variable	Frequency (Percent %)	Mean (SD)	Range
Age (years)		39.5 (±10.4)	20–64
Gender (*n* = 99)			
- Male	39 (39.4)		
- Female	60 (60.6)		
Co-morbidities (*n* = 99)			
- Nil known	79 (79.8)		
- Hypertension	10 (10.1)		
- HIV/AIDS	8 (8.1)		
- Diabetes	1 (1.01)		
- Asthma	1 (1.01)		
Mean IOP (mmHg)		14.4 (±2.6)	10–21

**Table 2 life-13-01979-t002:** Summary of the descriptive statistical analysis of the ONSD measurements 1 and 2 (taken in different planes at least thirty degrees apart), and the mean of the two readings for all the participants. All measurements were normally distributed as tested using Kolmogorov–Smirnov analysis.

Variable (mm)	Mean	SD	95% CI	Minimum	Maximum
ONSD 1	5.0	0.34	5.0–5.1	4.3	5.8
ONSD 2	5.1	0.34	5.0–5.1	4.3	6.0
Mean ONSD	5.1	0.33	5.0–5.2	4.4	5.9

## Data Availability

The data presented in this study are available on request from the corresponding author. The data are not publicly available due to privacy restrictions.

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
