# Peer review of "Optic Nerve Sheath Diameter: A Cross-Sectional Study of Ultrasonographic Measurement in Healthy Black South African Adults"

_life, 2023, doi:10.3390/life13101979_

Round 1
Reviewer 1 Report
This study reports normal range of ONSD in healthy black South African adults. The study does not give sufficiently sound information because of the following critical points.
Regression analysis is highly recommended in the determination of the factors associated with the ONSD. The authors' way to simply compare the ONSD difference according to each single parameter does not address the interaction between those parameters.
Given the subjectiveness of ONSD measurement, which is based on a manual measurement, it is inappropriate to use the ONSD measured from other study for comparison. An inter-observer, inter-measurement, and inter-device variability should be considered.
Author Response
Thank you very much for the feedback, my responses are listed below:
- Regression analysis is highly recommended in the determination of the factors associated with the ONSD. The authors' way to simply compare the ONSD difference according to each single parameter does not address the interaction between those parameters.
Regression analysis was done on the factors associated with ONSD as outlined in the statistical analysis section in lines 165-167 - "Independent t-test was used to test for differences across gender and co-morbidities and linear regression was used to assess the effect of age on ONSD, after correcting for gender and the effect of gender on ONSD after correcting for age. A one-sample t-test was used to compare our study finding of mean ONSD to the mean ONSD that was found in a Canadian population.[13]" The results for the regression analysis are given for age and gender in the results section lines 214-217. Our study was not powered to find statistical significance regarding the effect of co-morbidities on ONSD as this was not the main aim of our study and this is mentioned in the discussion as a limitation of the study.
Given the subjectiveness of ONSD measurement, which is based on a manual measurement, it is inappropriate to use the ONSD measured from other study for comparison. An inter-observer, inter-measurement, and inter-device variability should be considered
The aim of our study was to determine a normative database for the use of ONSD ultrasonography in our population group so that we can increase the accuracy of this tool which is already being used in our setting but with the average ONSD considered to be normal taken from international studies which we know is not accurate due to the factors mentioned in our discussion section. We did use interobserver analysis in our study by getting a second observer to review all measurements taken but we agree that the study could be improved by including more than one grader for the measurements and using different devices for the measurements (this is limited in our setting with the hospital using the same brand of ultrasound machine) and I have included that in the limitations and highlighted the change. There have been studies comparing the accuracy of ONSD on ultrasound to the measurement on CT/MRI scan measurements with good correlation but this was not the aim of our study.
Reviewer 2 Report
This manuscript demonstrated the measurement of the optic nerve sheath diameter in Black South African adults using ultrasound. It is interesting that the diameter is different compared with Canadian, but it is not surprising. The manuscript should address the issues as follows:
1. The p-value for the difference caused by sex and age is lower comparing with the presence of co-morbidities, do the authors still think these two factors have statistical significance?
2. The number of co-morbidities volunteers is small, which diminishes the statistical significance.
3. Recently, elastography is an emerging technology that is capable of detecting tissue stiffness non-invasively. The authors are encouraged to include the following works to expand the scope of this manuscript:
https://ieeexplore.ieee.org/document/9924584
https://doi.org/10.1109/TUFFC.2022.3190400
Author Response
Thank you very much for the review, the responses are mentioned below:
- The p-value for the difference caused by sex and age is lower comparing with the presence of co-morbidities, do the authors still think these two factors have statistical significance?
We feel that as mentioned in your second point, for true statistical significance of co-morbidities we would need a further study sufficiently powered with the aim to determine this and therefore we did not make any conclusion on statistical significance for the co-morbidities in this study as mentioned in the results and discussion. For gender and age, we had larger sample sizes and therefore the p value may be more significant than for co-morbidities, however it still did not have a P value <0.05. Other studies have also not found a correlation between gender and age on ONSD.
2. The number of co-morbidities volunteers is small, which diminishes the statistical significance.
We agree with this and have mentioned that our study was not powered to find statistical significance for this in the discussion (lines 270-272). It was not the main aim of our study and it would require a much larger sample size to assess for statistical significance regarding the correlation of co-morbidities on ONSD which we do feel would be a valuable inclusion for any further studies that we conduct on this.
3. Recently, elastography is an emerging technology that is capable of detecting tissue stiffness non-invasively. The authors are encouraged to include the following works to expand the scope of this manuscript.
Thank you for bringing this modality to our attention, although I could not find published papers on its use in optic nerve sheath measurements, I see how this could improve the accuracy of the traditional B scan approach and have included it in the discussion.
Reviewer 3 Report
In this manuscript, Trollip et al. assess the optic nerve sheath diameter (ONSD) in a healthy black population, by means of ultrasonography. This study is the first to report on ONSD in such population. However, it is limited by the manual grading performed by a single grader and by the fact that data collection is very limited (for example, they have not put findings in perspective with optical coherence tomography data).
1) Abstract, lines 15-16: This sentence can be removed.
2) Abstract, line 18: Just the mean? Or the distribution characteristics? Similarly in the aim paragraph of the Introduction, etc.
3) Abstract, lines 19-23: Too much repetition of “black South African population”.
4) Abstract, line 26: Please report only significant decimals and keep the number of decimals appearing in different places throughout the text and Tables consistent. In addition, please report as mean (SD) and remove the 95th percentile.
5) Introduction, line 44: What do the authors mean with the word ‘paired’? In addition, the optic nerve is more an extension of the retinal nerve fiber layer (axons), rather than the ganglion cell layer (bodies).
6) Methods, lines 122-126: Please stress that the measurements were manual and performed by a single grader.
7) Methods, lines 140-141: This is redundant. Figure 2 is also redundant, since the exclusion workflow is not too complex and can be understood easily just from the text.
8) Methods, lines 149-150: This study used a different ultrasound device. How are the authors sure that the results are comparable between devices, in order to make a claim about a real difference in the mean?
9) Methods, lines 151-152: This is redundant.
10) Results, lines 177-178 and Table 1: Why are these comorbidities of interest? They do not add to the manuscript.
11) Figure 3: Please add the x-axis.
Editing by a native speaker is required.
Author Response
Thank you for the feedback, the responses as follows:
In this manuscript, Trollip et al. assess the optic nerve sheath diameter (ONSD) in a healthy black population, by means of ultrasonography. This study is the first to report on ONSD in such population. However, it is limited by the manual grading performed by a single grader and by the fact that data collection is very limited (for example, they have not put findings in perspective with optical coherence tomography data).
Thank you for the feedback, although the vast majority of studies on ONSD measurements that we have analyzed did not include OCT analysis of any participants, we agree that the use of OCT would improve the study design and this is mentioned in the limitations as it was not feasible to do this on all our study participants due to the study being conducted in the emergency department where an OCT machine is not available, and therefore the decision was made to only include it on participants with higher cup-to-disc ratios as outlined in the method. The manual grading was done according to standardized procedure but we do agree that inter-observer grading would add value to studies on ONSD and this is mentioned as a limitation for our study.
1) Abstract, lines 15-16: This sentence can be removed.
Sentence has been removed from the abstract.
2) Abstract, line 18: Just the mean? Or the distribution characteristics? Similarly in the aim paragraph of the Introduction, etc.
The main aim of our study was to determine the mean ONSD in our population, for accurate use in a clinical setting. As a secondary objective we analysed the correlation of gender and age on ONSD and we have amended this in the article to clarify this further.
3) Abstract, lines 19-23: Too much repetition of “black South African population”.
This has been amended.
4) Abstract, line 26: Please report only significant decimals and keep the number of decimals appearing in different places throughout the text and Tables consistent. In addition, please report as mean (SD) and remove the 95th percentile.
This has been amended and the decimals of the mean rounded to two decimals.
This has been amended.
5) Introduction, line 44: What do the authors mean with the word ‘paired’? In addition, the optic nerve is more an extension of the retinal nerve fiber layer (axons), rather than the ganglion cell layer (bodies).
Paired was used to indicate that there are two optic nerves anatomically. this has been removed and amended to retinal nerve fibre layer.
6) Methods, lines 122-126: Please stress that the measurements were manual and performed by a single grader.
This has been amended.
7) Methods, lines 140-141: This is redundant. Figure 2 is also redundant, since the exclusion workflow is not too complex and can be understood easily just from the text.
These lines have been removed. Figure 2 has not been removed as requested by one of the authors, to provide an overview of the exclusion workflow.
8) Methods, lines 149-150: This study used a different ultrasound device. How are the authors sure that the results are comparable between devices, in order to make a claim about a real difference in the mean?
Thank you for mentioning this. We have mentioned this in the limitations with a suggestion at further studies considering inter-device variability. We have not been able to find a study comparing ONSD measured across different ultrasound machines and therefore cannot be sure that it will not account for the variation and this is now mentioned in the article.
9) Methods, lines 151-152: This is redundant.
This has been amended.
10) Results, lines 177-178 and Table 1: Why are these comorbidities of interest? They do not add to the manuscript.
These were all the co-morbidities reported by the study participants and were mentioned for transparency on factors that could influence ONSD as reported in other studies and mentioned in the discussion
11) Figure 3: Please add the x-axis.
The values of the x axis is written on the zero line but the plot has been enlarged and the values expanded for clarity. The amended plot is included in the revised manuscript.
Round 2
Reviewer 1 Report
The authors well addressed all the major concerns. Now I have minor points.
Abstract lines 24-25: Please describe how different the mean ONSD in the study was as compared to that in a Canadian population (larger or smaller).
Restulst lines 212-213: Remove "Linear regression analysis showed that age and gender account for 3% (R2=.030) of the variance in ONSD, (F(2,96)=1.465, p=0.236).", because the result is not meaningful without statistical significance.
Author Response
Thank you very much for the feedback...
Abstract lines 24-25: Please describe how different the mean ONSD in the study was as compared to that in a Canadian population (larger or smaller).
This has been amended
Restulst lines 212-213: Remove "Linear regression analysis showed that age and gender account for 3% (R2=.030) of the variance in ONSD, (F(2,96)=1.465, p=0.236).", because the result is not meaningful without statistical significance.
We feel that this is an important negative result as it is consistent with other studies published suggesting no statistically significant correlation between age and gender on ONSD. Linear regression analysis showed that in our sample, gender and age also did not correlate with ONSD, which agrees with the literature referenced in our article.
Reviewer 3 Report
The authors have addressed the minor points raised, but the manuscript is still limited by the lack of supporting data (such as structural OCT measurements and optic nerve grading verification). This is important and not merely a limitation, especially in the absence of a gold standard method, normative databases, and directly comparable literature (comparing measurements with a different study using a different device on a different population is not highly appreciated by the reviewer).
English editing is still required.
Author Response
Thank you for the response...
The authors have addressed the minor points raised, but the manuscript is still limited by the lack of supporting data (such as structural OCT measurements and optic nerve grading verification). This is important and not merely a limitation, especially in the absence of a gold standard method, normative databases, and directly comparable literature (comparing measurements with a different study using a different device on a different population is not highly appreciated by the reviewer).
We appreciate your suggestions, and will consider this for further studies as we cannot alter the study design for this study as data collection has been completed. ONSD is primarily used in the Emergency Department/Casualty setting and therefore the studies done on ultrasonographic ONSD that we have reviewed have not included any structural OCT measurements , fundoscopy, intraocular pressure measurement or optic nerve grading in their study design. Certainly in an ophthalmology clinic setting the use of OCT is an extremely useful tool to add to optic nerve assessment. Our main aim of this study is to determine the mean ONSD for our population which will aid us in using this tool in our casualty setting with the ultrasound machine used in our study, which has been determined to be 5.1mm, and this will allow us to use this for our patients going forward. In our analysis we compared it to the value of the Canadian population with the limitations mentioned in our article as you have pointed out in your previous review.
English editing is still required.
We are first language English speakers and have had the article reviewed for grammatical and spelling errors which have been amended. We have also requested review by the academic editor.